# Role of hydrogen bonding in hysteresis observed in sorption-induced swelling of soft nanoporous polymers

Mingyang Chen[1,2], Benoit Coasne [3], Robert Guyer[4], Dominique Derome[2] & Jan Carmeliet[1]

Hysteresis is observed in sorption-induced swelling in various soft nanoporous polymers. The associated coupling mechanism responsible for the observed sorption-induced swelling and associated hysteresis needs to be unraveled. Here we report a microscopic scenario for the molecular mechanism responsible for hysteresis in sorption-induced swelling in natural polymers such as cellulose using atom-scale simulation; moisture content and swelling exhibit hysteresis upon ad- and desorption but not swelling versus moisture content. Different hydrogen bond networks are examined; cellulose swells to form water–cellulose bonds upon adsorption but these bonds do not break upon desorption at the same chemical potential. These findings, which are supported by mechanical testing and cellulose textural assessment upon sorption, shed light on experimental observations for wood and other related materials.

[1] Chair of Building Physics, Department of Mechanical and Process Engineering, ETH Zurich, 8093 Zurich, Switzerland. [2] Laboratory for Multiscale Studies in Building Physics, Swiss Federal Laboratories for Materials Science and Technology, Ueberlandstrasse 129, 8600 Duebendorf, Switzerland. [3] CNRS, LIPhy, Univ. Grenoble Alpes, 38000 Grenoble, France. [4] Department of Physics, University of Nevada, Reno, 1664 N. Virginia Street, Reno, NV 89557, USA. Correspondence and requests for materials should be addressed to B.C. (email: benoit.coasne@univ-grenoble-alpes.fr) or to J.C. (email: cajan@ethz.ch)

Soft nanoporous matter is an important field which encompasses anthropic materials such as compliant porous solids, foams, intrinsically porous polymers, organic membranes, etc[1–3], as well as natural materials such as wood, bamboo, plants, linen, kerogen in gas shale, etc[4–8]. Owing to their large internal surface area and compliant skeleton, strong coupling between fluid configurations and deformation of these materials leads to sorption-induced swelling that can be accompanied with large hysteresis. For example, cellulose upon water adsorption swells by as much as 30%[9]. This swelling is accompanied by changes in the internal structure of the cellulosic system which modify the experience of the water molecules and feedback on the internal structure. This coupling between sorption and swelling in nanoporous materials has been investigated experimentally and with atomistic simulations[10–14] but also with phenomenological approaches such as the domain theory[15,16] (see ref [17]. for a recent review on water adsorption in wood). In parallel, much theoretical effort has been devoted to unify mechanical constitutive equations and surface thermodynamics relationships within the framework known as poromechanics[12,18–20]. In spite of being successful at describing the mutual effects of sorption and swelling, poromechanics provides no explanation for the microscopic mechanism of sorption-swelling coupling.

An important feature of natural nanoporous polymers is the hysteresis in sorption isotherms extending to very-low chemical potential[21–23]. This hysteresis phenomenon departs from that associated with capillary condensation, where hysteresis disappears below a certain chemical potential. There seems to be a consensus that sorption-induced swelling in such nanoporous polymers governs sorption hysteresis. This hypothesis still needs to be confirmed and the underlying microscopic mechanisms clarified (possible explanations cover dual mode adsorption theory[24] or hint to changes in hydrogen bonding configuration[25]). The purpose of this paper is to report an in-depth molecular simulation study that allows us to identify the details of the coupling mechanism responsible for the observed sorption-induced swelling and associated hysteresis.

At the molecular scale, despite its well-recognized ability to identify the molecular mechanisms driving sorption and transport in nanoporous media, molecular simulation based on statistical mechanics has failed so far to provide a unifying and comprehensive picture of hysteretic sorption-induced swelling in natural porous polymers. Monte Carlo simulations in the Grand Canonical ensemble (GCMC), by virtue of being at constant volume, is not suitable to the problem at hand[26]. Such GCMC simulations result in the absence of hysteresis illustrating that physical chemistry alone is not sufficient and deformation is therefore a necessary ingredient to achieve sorption hysteresis[27]. Molecular Dynamics (MD) allows considering the deformation of the host polymer, but assessing sorption isotherms by means of chemical potential estimation at given number of adsorbed molecules faces many technical issues (limited statistical accuracy, incomplete thermodynamic equilibration)[18]. By using a hybrid strategy combining GCMC and MD, the atomic simulations reported here probe adsorption phenomena while allowing for deformations and internal stress relaxation of the host porous polymer. Such microscopic simulations can explicitly account for the coupling between deformation of the porous material and water sorption.

Our simulation demonstrates that the hysteresis observed upon water sorption in cellulose-like materials stems from the coupling between the hydrogen bond network formed and the configuration/deformation of the host system. For a given moisture content, the host polymer exhibits, upon adsorption and desorption, different pore and energy landscapes that accommodate different hydrogen bond configurations (involving a ratio of water–water

to water–cellulose hydrogen bonds that differs between adsorption and desorption). While the first water molecules adsorb through hydrogen bonding at available sorption sites (intrinsic porosity of the dry material), subsequent water adsorption breaks intercellulose hydrogen bonds and induces swelling that exposes new hydrogen bonding sites at the cellulose chain surfaces. On desorption, the intercellulose hydrogen bonds, broken upon water adsorption, reform at a much lower relative humidity therefore leading to a large hysteresis.

## Results

**Sorption-induced swelling**. The molecular model of cellulose used in this work is shown in Fig. 1. As explained in the Methods section, our molecular models consist of a few cellulose chains made up of about 20 to 40 $\beta(1 \rightarrow 4)$ linked D-glucose units which are grown randomly (unit by unit) in an orthorhombic simulation box with periodic boundary conditions to avoid surface effects. The molecular model is then relaxed at a temperature $T = 300$ K and an external stress $\sigma = 0$ Pa in each direction—MD simulation in the isobaric-isothermal ensemble—to attain a density comparable to experimental values. Intramolecular interactions include bond stretching, bending, and torsion while intermolecular interactions consist of repulsion/dispersion contributions described using a Lennard-Jones potential combined with a Coulombic contribution. The parameters were taken from the PCFF (Polymer Consistent Force Field) forcefield[28] (see Supplementary Discussion for a discussion on the validity of the forcefield to describe cellulose). Three different molecular realizations, which were generated according to the strategy above, are used in the present paper to average data over a representative sample set. As shown in Supplementary Figure 4, these molecular realizations exhibit similar pore size distributions (PSD) with small pores < 2.5 Å only.

Water sorption in the cellulose material shown in Fig. 1 was investigated using molecular simulation in the $\mu\sigma T$ thermodynamic ensemble where the temperature $T$, the chemical potential $\mu$ for water and the external stress $\sigma$ are fixed. To do so, we used a hybrid strategy combining GCMC and MD in the isothermal-isostress

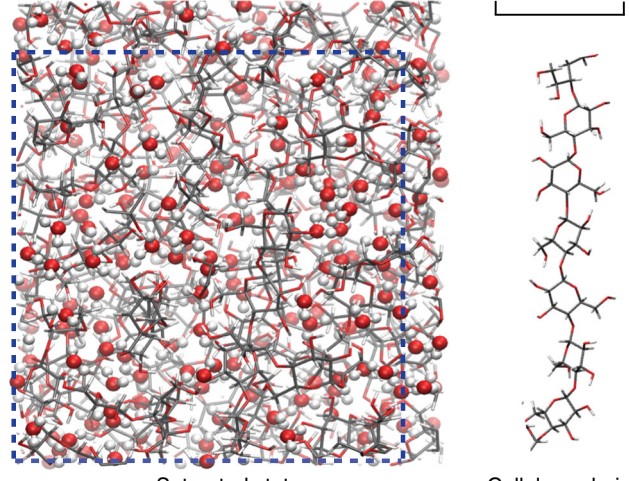

Saturated state                     Cellulose chain

**Fig. 1** Microscopic model of water sorption in cellulose. Molecular configuration of cellulose upon water adsorption at a relative humidity close to RH = 1 and a temperature $T = 300$ K. The sticks are the bonds between the C, O, and H atoms in cellulose while the red and white spheres are the O and H atoms of water. The box size is 3.10 × 3.31 nm. The blue dashed line shows the system size of the dry material before adsorption. A cellulose chain is shown on the right. The scale bar represents 1 nm

ensemble (MD-N$\sigma$T). While GCMC simulations allow probing water sorption in a system set in contact with an infinite reservoir imposing its chemical potential $\mu$ and temperature $T$, MD-N$\sigma$T simulations allow mechanical relaxation of the host matrix subjected to an external stress $\sigma$ at constant temperature $T$. Water is described using the SPC/E model (Extended Simple Point Charge model). Water–water and water–cellulose intermolecular interactions between the atoms include Coulombic, repulsion and dispersion contributions (with Lennard-Jones parameters determined using the Lorentz-Berthelot combining rules). Since $T$ is well below the critical point for water, its vapor is assumed to behave as an ideal gas $\mu \sim k_B T \ln P$ where $P$ is the vapor pressure, $T$ is the temperature, and $k_B$ is Boltzmann's constant. In practice, this means that, to simulate a given relative humidity $RH = P/P_0$ where $P_0 = 1017$ Pa is the bulk saturation vapor pressure for the SPC/E water model at $T = 300$ K, one imposes in the hybrid GCMC/MD simulations a chemical potential $\mu - \mu_0 \sim k_B T \ln RH$ ($\mu_0$ is the chemical potential of the water model at the bulk saturating vapor pressure $P_0$). Details of the procedure used to generate the molecular models and molecular interactions can be found in the Methods section while full details about the molecular simulations are provided in the Supplementary Method. The variations of system properties such as volume, stress, and adsorption amount when running to the equilibrium are given in Supplementary Figure 1, Supplementary Figure 2, and Supplementary Figure 3, respectively.

Figure 2 compares the simulated water adsorption/desorption isotherm in cellulose with its experimental counterpart[9]. These data, which show the water adsorbed amount $m$ as a function of the relative humidity RH, correspond to the average value over the three realizations (Supplementary Figure 5). Experimentally, cellulose is hard to obtain totally dehydrated prior to such water

adsorption experiments so that they contain a residual number of undesorbed water molecules. Moreover, such measurements as a function of RH suffer from the following limitation[9]. Even if sufficient time was allowed to reach thermodynamic equilibrium (including at RH = 0), measurement of the moisture content requires to take the sample out of the equilibration chamber so that it can be weighed. While this operation is done as fast as possible, some partial rehydration occurs. Consequently, for comparison, the experimental data were shifted up by $m = +0.05$ to account for the presence of residual i.e., non-desorbable water in the "dried" sample and unavoidable partial rehydration upon weighing. The shift $m = +0.05$ shows good agreement between the two datasets. In particular, when corrected for such a shift, the experimental data show typical moisture contents—including the maximum moisture content at RH = 1—and slopes in the adsorption/desorption isotherm that are consistent with the simulated data. The simulated sorption isotherms follow a trend very similar to the one of experimental data. The moisture content increases rapidly with RH in the low RH region and then increases less rapidly as initial pores in cellulose get filled. Upon further increasing RH, an inflection point in the adsorption isotherm is observed as pores open up upon swelling and significant subsequent adsorption occurs. Both the simulation and experimental data exhibit significant hysteretic behavior upon increasing/decreasing RH. Before addressing the microscopic origin of sorption-induced swelling hysteresis, it should be noted that possible mechanisms relying on chemistry (i.e., involving bond formation/breaking) cannot be envisaged as they would lead to irreversible sorption/swelling curves. If chemistry was involved in sorption-induced swelling, one would not recover the material properties upon decreasing the moisture content down to $m = 0$ (RH = 0). In contrast, in agreement with our molecular simulation data, all experiments on sorption-induced swelling of wood/cellulose show that one recovers the dry state upon dehydrating samples that have been first hydrated.

In order to probe the microscopic origin of hysteresis in sorption/swelling of cellulose, we performed additional simulations to test whether water adsorption/desorption cycles in undeformable cellulose lead to hysteresis. More precisely, we have simulated water adsorption/desorption in cellulose taken in its dry (unswollen) and fully hydrated (swollen) configurations. For these two states, we have performed a set of molecular simulations at constant volume in which the cellulose chains are allowed to relax (flexible) and a set of molecular simulations at constant volume in which the cellulose chains are kept frozen (frozen). Figure 2 shows the sorption isotherms obtained by simulating water ad- and desorption in cellulose when the model is maintained in its unswollen dry state and its swollen state at RH = 1. Adsorption/desorption cycles in the swollen and unswollen configurations are found to be hysteresis free (both in the flexible and frozen cellulose hosts). As expected, owing to the larger porous volume in swollen cellulose, the water adsorbed amount in the swollen state is larger than in the unswollen state for all RH. The different results above support the hypothesis that hysteretic behavior in sorption/swelling is governed by the coupling between sorption and swelling and not by a sorption or swelling alone.

The fact that hysteresis in such cellulose materials does not stem from the existence of different local minima in the fluid free energy is consistent with the absence of capillary hysteresis in nanoporous media with very small pores (typically < 2 nm)[29]. Indeed, for a material with pores of a size $D$, there is a capillary pseudocritical temperature $T_{cc}$ above which pore filling becomes reversible[30]. Statistical mechanics approaches, such as mean field theories and molecular simulations, but also experiments have shown that the shift between $T_{cc}$ and the bulk critical point $T_c$ is

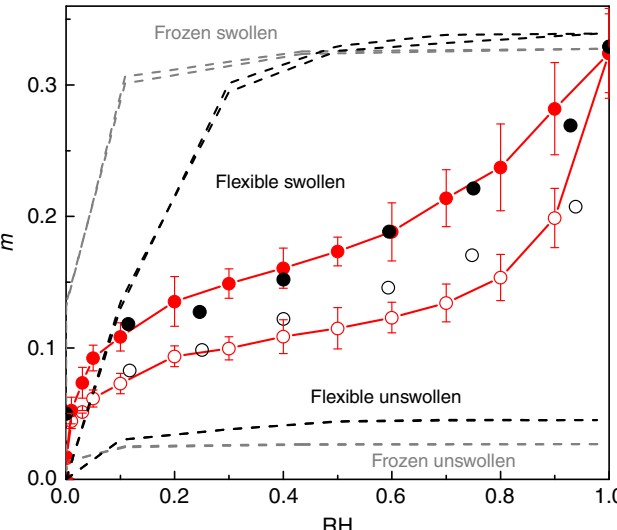

**Fig. 2** Water adsorption and desorption in cellulose. Water sorption isotherms at $T = 300$ K in cellulose: (red circles) hybrid GCMC/MD molecular simulation, (black circles) sorption experiment from ref [9]. The experimental data were shifted up by $m = +0.05$ to account for the presence of residual, i.e., non-desorbable water, in the dried material (see text). Open and closed symbols are adsorption and desorption data, respectively. The adsorbed amount is expressed as a moisture content $m$ defined as the mass of water per mass of dry material. The gray dashed lines correspond to the simulated, hysteresis-free sorption isotherms for water in the frozen unswollen and swollen cellulose material. The black dashed lines correspond to the same data but using simulations in which relaxation of the cellulose chains is allowed. The error bar is defined as the standard deviation (s.d.) of the three samples

such that $T_c - T_{cc} \sim 4\lambda T_c/D$ ($\lambda$ is the size of the confined molecule, $\lambda \sim 0.28$ nm for water)[21]. Considering the pore size in cellulose $D \sim 0.2$–$0.4$ nm, no capillary hysteresis should therefore be observed for cellulose-based materials such as wood (unless much larger pores are present).

Figure 3 shows the volume strain $\varepsilon_V$ as a function of relative humidity RH. The volume strain shows significant swelling with values as high as 36% close to RH = 1. The large hysteresis in sorption-induced swelling is also reflected in the strain data which show similar hysteretic behavior as the sorption isotherm. On the other hand, in agreement with experiments on wood[31], when plotted against moisture content $m$ (Fig. 3), the volume strains $\varepsilon_V$ upon adsorption and desorption collapse into a single linear relationship, $\varepsilon_V \sim m$. These results show that sorption-induced swelling is governed by the volume change induced upon adding new water molecules.

**Hysteresis and microscopic hydrogen bonding**. The data above establish that hysteresis observed in sorption-induced swelling arises from the coupling between sorption and deformation but they do not provide a microscopic picture of the underlying mechanisms. To examine the microscopic origin of such hysteresis in sorption-induced swelling, we carried out an in-depth analysis of the specific interactions involved upon water adsorption and desorption in cellulose. Amorphous cellulose is not a cross-linked polymer so that hydrogen bonds are expected to play a crucial role in its cohesion and mechanical behavior[32]. Herein a hydrogen bond is defined as a physical bond between an O and H atoms separated by a distance shorter than 0.35 nm and with an angle between the hydrogen donor and acceptor smaller than 30°[33,34]. There are three types of hydrogen bonds: bonds formed by cellulose groups with other cellulose groups ($HB^{CC}$), bonds formed by water with cellulose groups ($HB^{CW}$) and bonds formed by water molecules with other water molecules ($HB^{WW}$). While $HB^{WW}$, $HB^{CW}$, and $HB^{CC}$ provide important insights regarding the microscopic configuration of the water/chain upon cellulose hydration, we note that the fraction of water physisorbed—through hydrogen bonding—to the cellulose host represents a large fraction (as shown in Supplementary Figure 6, for all

moisture contents, at least 80% of the water molecules are hydrogen bonded to cellulose). Figure 4a, b shows $HB^{WW}$, $HB^{CW}$, and $HB^{CC}$ as a function of moisture content $m$ ($HB^{WW}$ and $HB^{CW}$ are normalized to the number of water molecules whereas $HB^{CC}$ is normalized to the number of cellulose groups). As expected, $HB^{CC}$ decreases with increasing the moisture content due to the breakage of cellulose intermolecular bonds upon swelling (the interchain distance increases so that fewer hydrogen bonds are formed). Concomitantly, more and more hydrogen bonds between water and cellulose form as more hydroxyl groups from cellulose become available upon swelling. Yet, $HB^{CW}$ decreases because it is normalized to the number of water molecules $\sim m$ which increases more rapidly than the number of cellulose–water hydrogen bonds. This interpretation is consistent with the increase in $HB^{WW}$ observed upon increasing $m$, which indicates that water clusters form inside cellulose pores.

While the different types of hydrogen bonds exhibit a marked hysteresis upon adsorption/desorption, Fig. 4a shows that the total number of hydrogen bonds formed by water $HB = HB^{CW} + HB^{WW}$ remains almost constant and close to the value in bulk water, $HB^{WW} \sim 3.6$[35]. This result suggests that the energetics of hydrated cellulose is governed by the formation of water–water and water–cellulose hydrogen bonds. As the number of water molecules increases with moisture content $m$, the cellulose microscopic structure evolves to accommodate water molecules and form sufficient hydrogen bonds. In fact, the criterion that water must form a number of HB close to the value occurring in liquid water can be taken as the signature of hydrophilic materials in the sense that the sorption energy is large enough to recover the cohesive energy in liquid water ($\sim 40$ kJ mol$^{-1}$).

The change in the microscopic structure of cellulose upon sorption manifests itself in the change in the PSD at different sorption states (Supplementary Figure 7). Starting from the PSD in the dry state, upon increasing $m$, bigger pores form to accommodate more water molecules so that fewer $HB^{CW}$ are observed while more $HB^{WW}$ form. In other words, as pores get bigger, their surface to volume ratio decreases so that $HB^{CW}$ decreases because they only form at the pore surface while $HB^{WW}$ increases because they form in the pore volume. More in detail, on the one hand, at low RH, water molecules adsorb on cellulose chains so that $HB^{CW}$ is large. On the other hand, as more water molecules get adsorbed upon increasing RH, they tend to form water clusters within the pores because of the limited number of available hydroxyl sites on the polymer chains. Meanwhile, the addition of water molecules makes the cellulose structure swell so that $HB^{CC}$ decreases (hydrogen bonds holding the cellulose chains together break upon swelling). By counting the total, unnormalized number of water–water ($n^{WW}$) hydrogen bonds, we found $n^{WW} \sim m^2$ which can be interpreted as follows. $HB^{WW}$ is readily obtained from the first peak in the OH pair distribution function $g(r)$ (the pronounced peak located at $\sim 1.9$ Å is characteristic of water–water hydrogen bonding) and the number of water–water hydrogen bonds for a single water molecule is defined as $\int 4\pi r^2 \rho g(r) \mathrm{d}r$ where the integral is over the range [$r_{min}$, $r_{max}$] where the HB peak is observed ($\rho$ is the water density). Since $g(r)$ is an intensive quantity, i.e., per water molecule, and assuming that $g(r)$ does not drastically change with moisture content, $n^{WW}$ is then obtained by multiplying by the number of water molecules in the system $N_w$. By noting that $\rho \sim m$ and $N_w \sim m$, we obtain $n^{WW} \sim m^2$.

A similar scenario holds upon desorption but differences in the water–cellulose and water–water energies coupled with different energy landscapes/host configurations upon adsorption and desorption makes the system follow different paths so that a small yet non negligible hysteresis is observed. To unravel the microscopic mechanism behind this hysteresis, typical energies

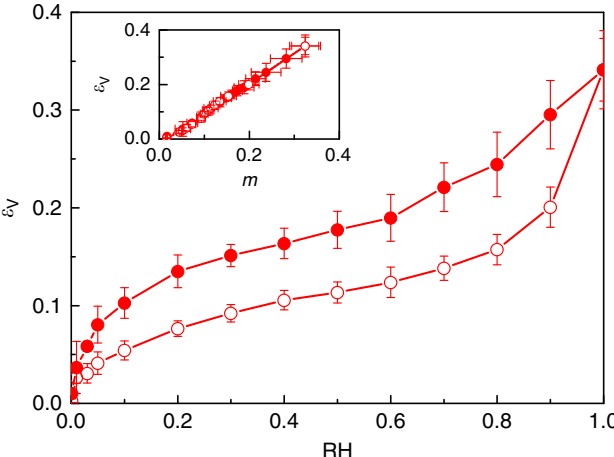

**Fig. 3** Sorption-induced swelling of cellulose. Volume strain $\varepsilon_V$ as a function of relative humidity RH upon water adsorption in cellulose at $T = 300$ K ($\varepsilon_V$ is defined as $V/V_0 - 1$ where $V$ and $V_0$ are the volumes of the hydrated and dry samples, respectively). The open and closed symbols, which correspond to adsorption and desorption, display a large hysteresis. The insert shows that the volume strain $\varepsilon_V$, plotted as a function of moisture content $m$, is not hysteretic. The error bar is defined as the standard deviation (s.d.) of the three samples

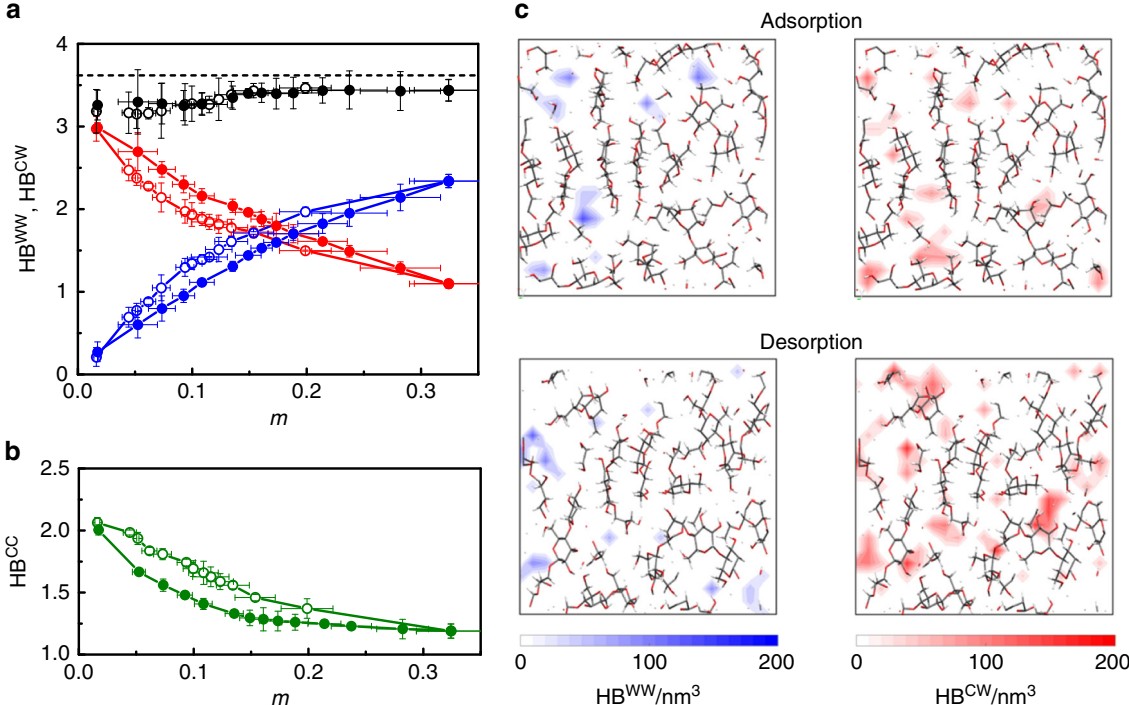

**Fig. 4** Hydrogen bonding mechanism inducing sorption hysteresis. **a** Number of hydrogen bonds per water molecule formed with other water molecules, $HB^{WW}$ (blue circles), and with cellulose, $HB^{CW}$ (red circles), as a function of moisture content $m$ for water adsorbed in cellulose. $HB^{CW}$ and $HB^{WW}$ are normalized to the number of water molecules. The black circles correspond to the total number of HB. Open and closed symbols correspond to adsorption and desorption, respectively. The dashed line indicates the number of HB per water molecule ~3.6 for bulk water at 300K[20]. **b** Number of hydrogen bonds per glucopyranose ring formed between cellulose, $HB^{CC}$ (green circles), as a function of $m$ for water adsorbed in cellulose. Open and closed symbols correspond to adsorption and desorption, respectively. **c** Density maps of $HB^{WW}$ (left, blue density scale) and $HB^{CW}$ (right, red density scale) upon water adsorption (top) and desorption (bottom) around a moisture content $m$ ~0.15. The error bar is defined as the standard deviation (s.d.) of the three samples

$E(HB^{CW})$ and $E(HB^{WW})$, corresponding to $HB^{CW}$ and $HB^{WW}$, were estimated by isolating hydrogen bonded groups from other molecules (i.e., other cellulose and water molecules). $E(HB^{CW})$ and $E(HB^{WW})$ were estimated by averaging over the total intermolecular energy (repulsive, dispersive, and coulombic contributions) between the two isolated groups (see Methods section). Independently of $m$ and the host configuration, we found that $E(HB^{WW}) \sim 4.2$ kcal mol$^{-1}$ is close to the value for bulk water while $E(HB^{CW}) \sim 5.4$ kcal mol$^{-1}$ is larger by ~1 kcal mol$^{-1}$. This result suggests that, as a $HB^{CW}$ is on average stronger than a $HB^{WW}$, removing water molecules from water clusters is more favorable than breaking water/cellulose bonds so that more water molecules remain attached to cellulose chains upon desorption. This interpretation is confirmed by HB density maps in Fig. 4c which show that, for the same moisture content $m$, the density/number of $HB^{CW}$ is larger upon desorption than adsorption (while the density/number of $HB^{WW}$ is smaller upon desorption than adsorption).

Sorption hysteresis also manifests itself in the mechanical properties of hydrated cellulose. Figure 5 shows the bulk modulus $K$ of hydrated cellulose as a function of the moisture content $m$. The bulk modulus in tension was extracted from the linear elastic part of the stress-strain relationship determined by a volumetric tensile test conducted using Molecular Dynamics at constant temperature $T$ in undrained conditions (see Methods section). $K$ decreases with increasing $m$, therefore displaying sorption-induced weakening. Weakening has been observed in other systems such as in rare gas adsorption in dense silica glasses[36]. Figure 5 shows that the bulk modulus $K$ at a given moisture content $m$ is lower upon desorption than adsorption but the hysteresis disappears when $K$ is plotted against the number of

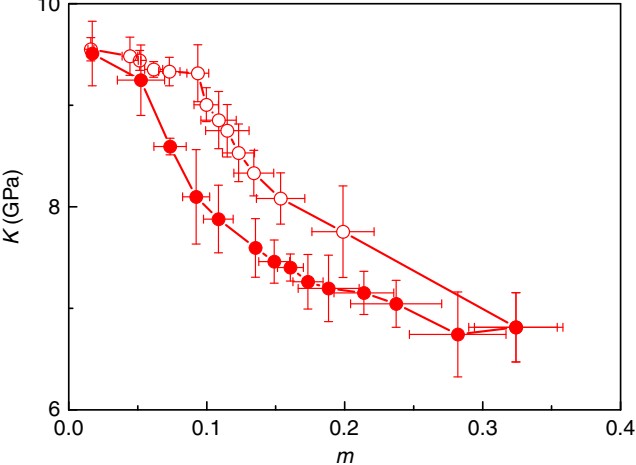

**Fig. 5** Mechanical properties of cellulose upon sorption. Bulk modulus $K$ as a function of moisture content $m$ upon water sorption in cellulose at $T = 300$ K. The open and closed symbols correspond to adsorption and desorption, respectively. The insert shows that the bulk modulus $K$, plotted as a function of the number of cellulose–cellulose hydrogen bonds $HB^{CC}$, is hysteresis-free. The error bar is defined as the standard deviation (s.d.) of the three samples

cellulose–cellulose hydrogen bonds, $HB^{CC}$. This result shows that the chain–chain intermolecular interactions govern the mechanical properties of hydrated cellulose. Comparing this result with the hysteresis-free behavior between moisture content and swelling (see insert Fig. 3) shows that sorption leads to different

behavior in sorption and mechanical loading. While we see a hysteresis upon mechanical loading when plotting the bulk modulus as a function of moisture content (due to hysteresis in hydrogen bonding), no hysteresis appears in swelling versus moisture content as it is driven by the pore volume change induced upon sorption and therefore linear with $m$.

## Discussion

Altogether, our results show that hysteresis observed in sorption-induced swelling of amorphous cellulose stems from different coupled microscopic hydrogen bond network/cellulose energy landscapes upon adsorption and desorption. This means that the hysteresis is not due to hydrogen bonding itself but to a complex coupling between the fluid grand free energy (that takes different values depending on the hydrogen bond network) and the cellulose free energy (which displays a rough energy landscape due to the complex pore network and cellulose conformation). The different paths followed by the system, which lead to the observed hysteretic behavior, arise from the existence of two distinct local minima in the hybrid thermodynamic potential corresponding to the sum of the water grand free energy and the cellulose free energy accounting for the deformation of the material. As a result, while adsorption and desorption at the same moisture content displays different water distributions, the hysteresis in the hydrogen bond network does not drive but reflects the observed hysteresis in sorption-induced swelling. Such hydrogen bond networks illustrate the different cellulose/water configurations in the thermodynamic ensemble corresponding to the so-called osmotic ensemble (where the corresponding thermodynamic potential i.e., fluid grand free energy + cellulose free energy has to be minimum).

From a configuration viewpoint, our results show that fewer cellulose–cellulose hydrogen bonds are formed in the wet state upon swelling and that different water–cellulose and water–water hydrogen bond distributions form at the same moisture content taken upon adsorption and desorption. Upon adsorption, cellulose–cellulose hydrogen bonds break due to swelling so that more hydroxyl sites are available to form water–cellulose hydrogen bonds. Upon desorption, due to the strong energy associated with cellulose–water hydrogen bonds, cellulose–cellulose hydrogen bonds do not reform at the same relative humidity than upon adsorption. Therefore, for a given moisture content $m$, the system accommodates the same number of water molecules but distributed according to different microscopic hydrogen bonding configurations corresponding to distinct host pore distributions/connectivities. This complex coupling leads to the hysteresis observed in experimental and simulated sorption isotherms for cellulose. While such mechanism was merely hypothesized in the literature, no experimental or theoretical evidence has been reported so far.

The insights gained in this paper show that hydrogen bonding also governs other important properties, such as the textural and mechanical properties of hydrated cellulose. The mechanism found in this article provides a clear microscopic understanding of the coupling between sorption and swelling in soft porous matter that goes well beyond wood and other cellulose-based materials. In particular, other soft porous materials such as kerogen in gas shale[37,38] but also man-made materials such as water artificial nanochannels[39] and polymer with intrinsic microporosity[40] are expected to display the same sorption/mechanical behavior. Extrapolation to wood should be made with caution as wood is a far more complex material than amorphous cellulose. Wood is a composite comprising different polymers in their amorphous and crystalline states.

## Methods

**Cellulose molecular model**. The molecular model of dry amorphous cellulose was constructed with the help of Materials Studio 8.0. While cellulose can be obtained from various sources (cotton cellulose, bacterial cellulose, etc.), we considered cellobiose as it corresponds to the base unit of higher plants. Cellobiose, a reducing sugar, is a disaccharide with the formula $C_{12}H_{22}O_{11}$ and consists of two β-glucose molecules linked by a β(1 → 4) bond. Cellobiose was firstly built as the repeat unit, after which the initial configuration of the cellulose chain was generated using a Monte Carlo method and then packed into a periodic orthogonal box by employing the Amorphous Cell Module at an initial density of 1.2 g cm$^{-3}$. For better statistics, three samples with different chain numbers and chain lengths were prepared as listed in Supplementary Table 1. The initial configurations of the three samples were imported into LAMMPS (Large-Scale Atomic/Molecular Massively Parallel Simulator)[41] for relaxation using Molecular Dynamics simulation at constant temperature/stress with a Nose-Hoover thermostat/barostat (NσT)[42]. The temperature and stress were set to 300 K and 0 Pa, respectively. The PCFF forcefield[28] was employed to describe both intramolecular and intermolecular interactions. The long-range electrostatic interactions were implemented using the Ewald summation method. Periodic boundary conditions were imposed in the three directions to avoid size effects. Molecular dynamics was carried out for the three samples for 2 ns with an integration time step and thermostat relaxation time being 1 fs and 500 fs, respectively. We found that both energy and density converged after 2 ns. The final molecular configurations for each sample are shown in Supplementary Figure 4. As listed in Supplemental Table 1, the final densities were consistent with the experimental density. The PSD of the three samples, which are shown in Supplementary Figure 4, were calculated using a constrained, nonlinear optimization method described in ref [43]. (using a van der Waals radius = 0.1 nm for PSD calculations).

**Molecular simulation of cellulose adsorption/desorption**. Using the three samples presented above, water adsorption and desorption in deformable cellulose was simulated at room temperature by combining Grand Canonical Monte Carlo and Molecular Dynamics simulations. For the adsorption branch, the dry amorphous cellulose samples were selected as the initial configuration and molecular simulations at different chemical potentials were conducted to obtain the adsorbed amount (moisture content, $m$) as a function of water relative humidity RH. Then, starting from the state close to RH = 1, desorption was simulated by decreasing the water relative humidity (which is related to the water chemical potential). Full details about the molecular simulation techniques can be found in the Supplementary Method. The simulated adsorption/desorption isotherms for the three cellulose samples are shown in Supplementary Figure 5.

**Hydrogen bond energies**. Hydrogen bond energies in our simulation were determined based on post-process analysis of the different configurations generated in the course of the molecular simulation. For each hydrogen bond, we isolated the two molecules that are hydrogen bonded by removing all the remaining molecules. Then, we calculated the total intermolecular energy between the two molecules including both Lennard-Jones and Coulombic contributions. By averaging over the trajectory, the binding energies corresponding to HB$^{WW}$ and HB$^{CW}$ can be determined.

**Mechanical testing**. The tensile stress-strain curve for each cellulose model was measured under a linear volumetric strain protocol where the volumetric strain is increased from $\varepsilon_V = 0$ to 0.1 at constant temperature $T = 300$ K. The volumetric strain $\varepsilon_V$ was imposed by changing the volume of the simulation box in the frame of a dynamics run. Such a tensile process was conducted within a time window of 4 ns, which corresponds to a strain rate low enough so that strain rate effects can be neglected. The resulting tensile stress $\sigma$ was extracted in a continuous fashion during the loading procedure to obtain the stress-strain curve shown in Supplementary Figure 8. By determining the slope at small strains, i.e., in the linear/elastic regime, the bulk modulus $K$ of the hydrated material was determined. Note that loading was conducted under MD simulation (at constant moisture content) so that $K$ corresponds to the undrained bulk modulus.

## Data availability

The molecular structural data of dry and hydrated cellulose that support the findings of this study are available in the public repository 'ETH Research Collection'.[44]

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

## Acknowledgements

The authors acknowledge the support of the Swiss National Science Foundation (SNSF) (No.143601). D.D. acknowledges Yves Fortin for having in the past inspired this investigation.

## Author contributions

D.D. and J.C. conceived and directed the research; M.C. determined the simulation methodology with the help of B.C.; M.C. performed all simulations; M.C. performed the post-processing of results with the help of B.C. and R.G; and all authors analyzed the results, contributed to the scientific discussions, and the preparation and writing of the manuscript.

## Additional information

**Competing interests:** The authors declare no competing interests.

