## [Peer Review File · Nature Communications]

Reviewers' comments:

Reviewer #1 (Remarks to the Author):

This is an interesting and novel piece of work on the role of hydrogen bonding in hysteresis observed in cellulose. The analysis is scientifically sound and quite elegant. The manuscript is worth publishing after the authors address the concerns and recommendations of this reviewer as described below.

1) The authors should specify the simulated cellulose material, namely, what kind of cellulose were they simulating? Cellulose from various pulping source? Cotton? Bacterial cellulose?

2) The authors claimed they have built an atomic scale cellulose model, but with no verification from experiments. If the cellulose is partly crystallised (in most types of cellulose), they could verify the simulated cellulose crystal structure (crystallinity, lattice size etc.) using the X-ray/neutron diffraction data. In other words, does the microfibril they constructed make sense? Only when the simulated cellulose structure passes this initial test, we can talk about others, either sorption or mechanical properties.

3) The GCMC+MD approach was used to address the swelling occurred in the sorption process, but the essential parts were not provided. The external stress they used in MD seems to be important in the entire simulation, but no calculation method was provided.

4) How did they run the GCMC and MD for a specified relative humidity (or chemical potential)? Run GCMC and MD alternatively for a differential volume increase? Did they use the MD once to obtain a fixed volume and then use the GCMC to obtain the quantity of adsorbed water molecules?

4) Their arguments on hysteresis based on hydrogen bonding are not fully convincing, Observed different number of HB for desorption and adsorption can be the consequences of other causes like metastable states of bound water or pore connectivity, not necessarily direct causes. Furthermore, HB's role on sorption was questioned by Hill et al. experimentally. More attention to this kind of experimental evidence should be paid.

5) The simulated bound water structure is questionable. It seems that they only observed water clusters. The lower density of water clusters is not consistent with the measured bound water density; and the faster mobility of water clusters compared to liquid bulk water is against the slower mobility from NMR tests. My guess is that the simulated cellulose structure is too soft, which made the micropores more spherical rather than the long slit or cylindrical pores.

6) The authors should not extrapolate to the case of wood. Lignin's important role on wood has been experimentally proven by Hill et al. and Christensen. The authors should be aware that conclusions on cellulose cannot be directly applied on wood.

7) Other similar works by Shi and Avramidis are not included in the manuscript. Please add to references the following papers:

Shi, J. and S. Avramidis. 2017. Water sorption hysteresis in wood: III physical modeling by molecular simulation. *Holzforschung*, DOI 10.1515/hf-2016-0231.

Shi, J. and Avramidis, S. 2017. Water sorption hysteresis in wood: II mathematical modeling – functions beyond data fitting. *Holzforschung*, 71(4): 317-321.

Shi, J. and Avramidis, S. 2017. Water sorption hysteresis in wood: I review and experimental patterns – geometric characteristics of scanning curves. *Holzforschung*, 71(4): 307-316.

My recommendation is "publish after corrections".

Reviewer #2 (Remarks to the Author):

The authors reported a hybrid scheme of GCMC and MD simulations on sorption-induced swelling using molecular model of cellulose. They tried to propose a solution on the long standing question--why moisture content and swelling exhibit hysteresis upon ad- and desorption but not swelling versus moisture content.

My first impression after reading the manuscript was that the authors had overrated their results. I admit that there are some interesting output from this study. However, I don't think the manuscript meets the criterion of Nat. Commun. At the moment, in the present form, the manuscript is misleading and some descriptions are inadequate. Specifically:

1. As stated in the title and Abstract, the authors demonstrated that the role of hydrogen bonding is the molecular mechanism responsible for the hysteresis in sorption-induced swelling in natural polymers. However, my general comment is that one has to rule out other possibilities to confirm that something is responsible. Yes, results shown in Fig. 4 seem to be a solution. However, in my opinion, a strong connection is missing. I think what I found in this manuscript "Amorphous cellulose is not a cross-linked polymer so that hydrogen bonds are expected to play a crucial role in its cohesion and mechanical behavior" is not enough.

2. I do not question the reliability of the simulation, but it is well-known that the accuracy of the classical force field is a big problem. For the hybrid strategy combining GCMC and MD, has it been verified? Is there any validation?

3. Concerning the results and their interpretation, some treatments are clearly lack of strong proof. For example, lines 138-139, "For the sake of comparison, the experimental data were shifted up by $m = +0.05$ to account for the presence of residual i.e. non-desorbable water in the 'dried' sample." It sounds reasonable but I rather believe that it should be much more complex. Why a shift of 0.05, not any other possible value? Is the presence of residual the only reason resulting the difference?

Minor points.

1. Terminology of MD simulations is not consistent through the entire manuscript, such as isothermal-isobaric ensemble (MD-N σ T) in line 115 [then σ was used as the size of the confined molecule (line 161)], however in the Methods section, it becomes to "constant temperature/pressure with a Nose-Hoover thermostat/barostat (NPT)" in line 353.

2. I am not sure whether all the simulation can be reproduced with the given poor description, especially for the GCMC part, regarding to change the RH.

Dr Benoit Coasne
Tel: +33 6 70 80 12 34
benoit.coasne@univ-grenoble-alpes.fr

Laboratoire Interdisciplinaire de Physique
140 Av. de la physique
38000 Grenoble

Grenoble, May 8, 2018

* * * * *

Reviewer #1

This is an interesting and novel piece of work on the role of hydrogen bonding in hysteresis observed in cellulose. The analysis is scientifically sound and quite elegant. The manuscript is worth publishing after the authors address the concerns and recommendations of this reviewer as described below. My recommendation is "publish after corrections".

Comment 1. *The authors should specify the simulated cellulose material, namely, what kind of cellulose were they simulating? Cellulose from various pulping source? Cotton? Bacterial cellulose?*

Reply 1. We agree with the reviewer that such information should be specified clearly. In order to address this comment, we have rephrased the following sentences in the revised version of the paper (Page 17): “The molecular model of dry amorphous cellulose was constructed with the

help of Materials Studio 8.0. While cellulose can be obtained from various sources (cotton cellulose, bacterial cellulose, etc.), we considered cellobiose as it corresponds to the base unit of higher plants. Cellobiose, a reducing sugar, is a disaccharide with the formula $C_{12}H_{22}O_{11}$ and consists of two β -glucose molecules linked by a $\beta(1\rightarrow4)$ bond. Cellobiose was firstly built as the repeat unit, after which the initial configuration of the cellulose chain was generated using a Monte Carlo method and then packed into a periodic orthogonal box by employing the Amorphous Cell Module at an initial density of 1.2 g/cm^3 . For better statistics, three samples with different chain numbers and chain lengths were prepared as listed in Supplementary Table 1.”

Comment 2. *The authors claimed they have built an atomic scale cellulose model, but with no verification from experiments. If the cellulose is partly crystallised (in most types of cellulose), they could verify the simulated cellulose crystal structure (crystallinity, lattice size etc.) using the X-ray/neutron diffraction data. In other words, does the microfibril they constructed make sense? Only when the simulated cellulose structure passes this initial test, we can talk about others, either sorption or mechanical properties.*

Reply 2. Before addressing in details the validity of our cellulose model, we would like to comment on the cellulose type considered in the present work. As detailed in our reply to the previous comment, we model cellobiose that corresponds to the base unit in higher plants. Consequently, we do not claim to build a microfibril which is a far more complex system having more than one polymer and where about 50% of the cellulose is known to be crystalline [for these different systems, we refer to the previous work by Kulasinski, Derome, and Carmeliet in *Sci. Rep.* (2017); Kulasinski, Guyer, Derome, Carmeliet in *Biomacromol.* (2015)]. Here, in contrast, we are interested in investigating one component of the cell wall layer in a completely amorphous state. We understand from the reviewer’s comment that this point should be made clearer; We have added the following information in the revised version of the manuscript (Page 17): “On the other hand, extrapolation to wood should be made with caution as wood is a far more complex material than amorphous cellulose since wood consists of a composite made of different polymers in their amorphous and crystalline states.”

Regarding the validity of the cellobiose model used in our work, we agree with the reviewer that the initial manuscript did not discuss carefully the validity of the forcefield used to describe cellulose. In order to address this important point, we have added in the revised version of our Supplementary Information a section dedicated to the discussion on the force field validity (SI file, Page 3):

“II. Discussion on the forcefield validity to describe cellulose

PCFF is a force field parameterized against a broad range of experimental observables for organic compounds. It has been applied to modelling cellulose-based materials by many researchers who have found that it captures quantitatively or semi-quantitatively most physical properties. More in details, for instance, Chen et al. [2] built an

amorphous cellulose model and showed that the final density obtained is consistent with its experimental counterpart (1.39 g/cm^3 versus 1.48 g/cm^3). Similarly, in the present work, we found that the final density of our models (ranging from 1.39 to 1.41 g/cm^3) is consistent with typical experimental densities for cellulose (see Supplementary Table 1). As far as mechanical properties are concerned, Tanaka and Iwata [3] used molecular simulation with the same force field to assess Young's modulus of cellulose crystals; these authors found values in the range 124 - 155 GPa that are in good agreement with the experimental value $\sim 138 \text{ GPa}$ [4] (no experimental mechanical data are available for amorphous cellulose). As for adsorption properties, in their molecular simulation of adsorption onto cellulose, Da Silva Perez et al. [5] found that the heat of adsorption for a large variety of aromatic compounds is consistent with their experimental counterpart (84% of the adsorbate-cellulose couples displayed differences $< 20\%$ between the measured and predicted heats of adsorption). Xu and Chen [6] also found that PCFF predicts formaldehyde diffusion in cellulose with a temperature dependence of the self-diffusion coefficient in good agreement with the experimental data. Finally, in the context of the present work on water adsorption/desorption in cellulose, we emphasize that PCFF leads to cellulose/water hydrogen bonds with a typical energy (5.4 kcal/mol , see discussion in our manuscript) that is consistent with the conformational analysis made by Pizzi et al. [7,8]; these authors estimated theoretically that the sorption energy is around 5.5 kcal/mol for cellulose I crystals and 6.5 kcal/mol for paracrystalline (amorphous) cellulose. Overall, the discussion above shows that the forcefield used in the present work provides a reasonable, at least semi-quantitative, description of cellulose (including its density, mechanical, and adsorption properties).”

Comment 3. *The GCMC+MD approach was used to address the swelling occurred in the sorption process, but the essential parts were not provided. The external stress they used in MD seems to be important in the entire simulation, but no calculation method was provided.*

Reply 3. We agree with the reviewer that many technical yet essential elements of the computational approach used in our work are missing. In order to address this comment, we have added in the revised version of our manuscript the following elements.

- We have added in the Supplementary Information a section dedicated to the computational methods (Page 3):

“I. Combining Grand Canonical Monte Carlo and Molecular Dynamics

As explained in our paper, state-of-the-art techniques such as Molecular Dynamics (MD) and Grand Canonical Monte Carlo (GCMC) do not allow describing coupled adsorption/swelling in porous materials. Indeed, while the former requires to work

with a constant number of molecules, the latter only applies to systems having a constant volume. Following previous works, including the work by Ghoufi and Maurin for adsorption in Metal Organic Frameworks [1], we use a hybrid molecular simulation that consists of combining GCMC and MD to perform simulations in the so-called osmotic statistical ensemble $\mu\sigma T$ where μ is the chemical potential of water, σ is the stress applied to the cellulose material, and T the temperature. To do so, part of the Markov chain used in the Grand Canonical Monte Carlo simulations is replaced by an MD trajectory at constant number of water molecules N , constant stress σ and temperature T . More in details, this means that, in addition to conventional MC steps in the GCMC algorithm (insertions and deletions), MD timesteps *i.e.* at constant number of molecules (allowing for water molecule translations or rotations and cellulose local relaxation at constant external stress) are added as the MD technique is more efficient at converging towards local equilibrium. Moreover, by using MD simulations in the isostress and isothermal ensemble, the host (hydrated) cellulose material is allowed to swell or shrink. As shown in previous works, such a hybrid strategy succeeds in capturing coupled adsorption/swelling phenomena including adsorption-induced phase transitions in breathing materials such as Metal Organic Frameworks [1].

In practice, our hybrid molecular simulations were performed at $T = 300$ K using a Berendsen thermostat and an anisotropic external stress $\sigma = 0$ Pa (with the following relaxation times: $\tau_T = 0.1$ ps and $\tau_s = 1$ ps). Water molecules are described using the SPC/E water model with the SHAKE algorithm to maintain its internal structure rigid. The MD trajectory was integrated using the velocity Verlet integrator with a timestep equal to 1 fs. Hybrid MD/GCMC molecular simulations consisted of performing a large number of blocks where one block corresponds to 2000 GCMC insertion/deletion attempts followed by 200 MD timesteps. In total, 10^5 blocks were first performed to equilibrate the system followed by 2×10^4 additional blocks to accumulate statistics.”

- For the sake of clarity, we have also added the following sentences in the revised version of the manuscript (Page 5): “Details of the procedure used to generate the molecular models and molecular interactions can be found in the Methods section while full details about the molecular simulations are provided in Section I of the Supplementary Information.” and (Page 18): “Full details about the molecular simulation techniques can be found in the Supplementary Information in which a section is dedicated to the computational methods.”

Comment 4. *How did they run the GCMC and MD for a specified relative humidity (or chemical potential)? Run GCMC and MD alternatively for a differential volume increase? Did*

they use the MD once to obtain a fixed volume and then use the GCMC to obtain the quantity of adsorbed water molecules?

Reply 4. Once again, we agree with the reviewer that these technical elements are missing. While such important information has been discussed in the previous point, we report in this response specific details to answer this question from the reviewer. In particular, we emphasize that the combination of GCMC and MD relies on a robust strategy in which one samples, as in the corresponding experiments, the appropriate ensemble where the chemical potential μ , external stress applied to the system σ , and temperature T are constant. This implies that, as in experiments on wood swelling upon sorption, the number N of water molecules is allowed to vary while the stress and temperature applied to the sample are controlled. In practice, as described in our response to the previous comment by the same reviewer, GCMC and MD are combined in such a way to sample the appropriate ensemble known as the osmotic ensemble $\mu\sigma T$ (and not simply by running an MD run followed by a GCMC run or vice versa). In order to address this comment, all details are now described in a paragraph that has been added in the revised version of the Supplementary Information (SI file, Page 2).

Comment 5. *Their arguments on hysteresis based on hydrogen bonding are not fully convincing. Observed different number of HB for desorption and adsorption can be the consequences of other causes like metastable states of bound water or pore connectivity, not necessarily direct causes. Furthermore, HB's role on sorption was questioned by Hill et al. experimentally. More attention to this kind of experimental evidence should be paid.*

Reply 5. We agree with the reviewer; we do not claim that hysteresis is directly caused by hydrogen bonding itself but that it is due to a coupling between the fluid grand free energy (that takes different values depending on its hydrogen bond network) and the cellulose free energy (which displays a rough energy landscape due to the complex pore network and cellulose conformation). As a result, while adsorption and desorption at the same moisture content displays different water distributions, the hysteresis in the hydrogen bond network does not drive but reflects the observed hysteresis in sorption-induced swelling. Such hydrogen bond networks reflect the different cellulose/water configurations in the thermodynamic ensemble corresponding to the so-called osmotic ensemble where the corresponding thermodynamic potential (fluid grand free energy + cellulose free energy) has to be minimum. From a configuration viewpoint, our results show that, upon swelling, fewer cellulose-cellulose hydrogen bonds form in the wet state and different water-cellulose and cellulose-water hydrogen bonds distributions form at the same moisture content taken upon adsorption and desorption. We understand from the reviewer's comment that our conclusion/discussion is misleading. In order to address this comment, we have rephrased the following paragraph (Pages 15-16): "**This means that the hysteresis is not due to hydrogen bonding itself but to a complex coupling between the fluid grand free energy (that takes different values depending on the hydrogen bond network) and the cellulose free energy (which displays a rough energy landscape due to the complex pore network and cellulose conformation). The different paths followed by the system, which lead to the observed hysteretic behavior, arise from the existence of two distinct local minima in the hybrid thermodynamic potential corresponding to the sum of the water grand free energy and the cellulose free energy accounting for the deformation of the material. As a result, while adsorption and desorption at the same moisture content displays different water distributions, the**

hysteresis in the hydrogen bond network does not drive but reflects the observed hysteresis in sorption-induced swelling. Such hydrogen bond networks illustrate the different cellulose/water configurations in the thermodynamic ensemble corresponding to the so-called osmotic ensemble (where the corresponding thermodynamic potential i.e. fluid grand free energy + cellulose free energy has to be minimum).

From a configuration viewpoint, our results show that fewer cellulose-cellulose hydrogen bonds are formed in the wet state upon swelling and that different water-cellulose and water-water hydrogen bond distributions form at the same moisture content taken upon adsorption and desorption. More in details, upon adsorption, cellulose-cellulose hydrogen bonds break due to swelling so that more hydroxyl sites are available to form water-cellulose hydrogen bonds. On the other hand, upon desorption, due to the strong energy associated with cellulose-water hydrogen bonds, cellulose-cellulose hydrogen bonds do not reform at the same relative humidity than upon adsorption. Therefore, for a given moisture content m , the system accommodates the same number of water molecules but distributed according to different microscopic hydrogen bonding configurations corresponding to distinct host pore distributions/connectivities. This complex coupling leads to the hysteresis observed in experimental and simulated sorption isotherms for cellulose. While such mechanism was merely hypothesized in the literature, no experimental or theoretical evidence has been reported so far.”

Comment 6. *The simulated bound water structure is questionable. It seems that they only observed water clusters. The lower density of water clusters is not consistent with the measured bound water density; and the faster mobility of water clusters compared to liquid bulk water is against the slower mobility from NMR tests. My guess is that the simulated cellulose structure is too soft, which made the micropores more spherical rather than the long slit or cylindrical pores.*

Reply 6. We do not fully understand the reviewer's comment as we do not observe water clusters. In fact, quite the opposite, we observe that a large fraction of water are physisorbed – through hydrogen bonding – to the cellulose host (for all moisture contents, we found that at least 80% of the water molecules are hydrogen bonded to cellulose – see Figure S3 in the Supplementary Information file). It is not the purpose of this paper to look in detail at the dynamics of water in cellulose; however considering that the water-cellulose hydrogen bond energy is larger than the water-water hydrogen bond energy, we anticipate that the self-diffusivity of adsorbed water is slower than that the other water molecules and bulk water molecules (in agreement with the experiments mentioned by the reviewer). Regarding the cellulose structure, it should be emphasized that the mechanical Young modulus of the cellulose model as described using the forcefield PCFF, is in very good agreement with its experimental counterpart. We understand from the reviewer's comment that additional information

should be given. In order to address this comment, we have made the two following modifications:

- Regarding the mechanical properties of cellulose, we have added the following sentences (see Section II in the Supplementary Information): “As far as mechanical properties are concerned, Tanaka and Iwata [3] used molecular simulation with the same forcefield to assess Young’s Modulus of cellulose crystals; these authors found values in the range 124-155 GPa that are in good agreement with the experimental value (~138 Gpa) [4] (no experimental mechanical data are available for amorphous cellulose).”
- Regarding the fraction of water physisorbed to the cellulose host, we have added the following sentence in the revised version of the manuscript (Page 11): “While HB^{WW} , HB^{CW} and HB^{CC} provide important insights regarding the microscopic configuration of the water/chain upon cellulose hydration, we note that the fraction of water physisorbed – through hydrogen bonding – to the cellulose host represents a large fraction (as shown in Supplementary Fig. S3, for all moisture contents, at least 80% of the water molecules are hydrogen-bonded to cellulose).”
- We have also added for completeness in the revised Supplementary Information file the following figure:

Supplementary Figure S3. Fraction of water molecules hydrogen-bonded to the cellulose chains. Open and closed symbols correspond to adsorption and desorption, respectively.

Comment 7. *The authors should not extrapolate to the case of wood. Lignin's important role on wood has been experimentally proven by Hill et al. and Christensen. The authors should be aware that conclusions on cellulose cannot be directly applied on wood.*

Reply 7. We agree that extrapolation to wood should be discussed with caution. We understand from the reviewer's comment that our paper is misleading in its current version. In order to address this comment, we have rewritten the discussion of our paper to tone down our conclusions (note that we also remove the statement "*and wood in general*" in the sentence "*This complex coupling leads to the hysteresis observed in experimental and simulated sorption isotherms for cellulose, and wood in general*". The revised version of our discussion/conclusion reads (Page 17): "**On the other hand, extrapolation to wood should be made with caution as wood is a far more complex material than amorphous cellulose since wood consists of a composite made of different polymers in their amorphous and crystalline states.**"

Comment 8. *Other similar works by Shi and Avramidis are not included in the manuscript. Please add to references the following papers: Shi, J. and S. Avramidis. 2017. Water sorption hysteresis in wood: III physical modeling by molecular simulation. Holzforschung, DOI 10.1515/hf-2016-0231. Shi, J. and Avramidis, S. 2017. Water sorption hysteresis in wood: II mathematical modeling – functions beyond data fitting. Holzforschung, 71(4): 317-321. Shi, J. and Avramidis, S. 2017. Water sorption hysteresis in wood: I review and experimental patterns – geometric characteristics of scanning curves. Holzforschung, 71(4): 307-316.*

Reply 8. We are grateful to the reviewer for bringing to our attention these interesting papers which are indeed relevant to the present work. They have been added/discussed in the revised introduction of our manuscript (Page 2): "**This coupling between sorption and swelling in nanoporous materials (including wood) has been investigated experimentally and with atomistic simulations¹⁰⁻¹⁴ but also with phenomenological approaches such as the domain theory¹⁵⁻¹⁶ (see Ref. 17 for a recent review on water adsorption in wood).**"

* * * * *

Reviewer #2

The authors reported a hybrid scheme of GCMC and MD simulations on sorption-induced swelling using molecular model of cellulose. They tried to propose a solution on the long standing question---why moisture content and swelling exhibit hysteresis upon ad- and desorption but not swelling versus moisture content. My first impression after reading the manuscript was that the authors had overrated their results. I admit that there are some interesting output from this study. However, I don't think the manuscript meets the criterion of Nat. Commun. At the moment, in the present form, the manuscript is misleading and some descriptions are inadequate.

Comment 1. *As stated in the title and Abstract, the authors demonstrated that the role of*

hydrogen bonding is the molecular mechanism responsible for the hysteresis in sorption-induced swelling in natural polymers. However, my general comment is that one has to rule out other possibilities to confirm that something is responsible. Yes, results shown in Fig. 4 seem to be a solution. However, in my opinion, a strong connection is missing. I think what I found in this manuscript “Amorphous cellulose is not a cross-linked polymer so that hydrogen bonds are expected to play a crucial role in its cohesion and mechanical behavior” is not enough.

Reply 1. We agree with the reviewer that it is important to verify that other possible scenarios do not explain adsorption-induced swelling in cellulose materials. In order to address this important point, we have carried out additional simulations to rule out other possibilities in which hysteresis would be due to sorption effects without swelling. In other words, other possibilities refer here to any adsorption mechanism such as capillary condensation, hysteresis in wetting properties (dynamic contact angle), hysteretic local deformation/relaxation, etc. Before going into the details of the additional work that has been carried out, we emphasize that mechanisms relying on chemistry (i.e. involving bond formation/breaking) are to be excluded as they would lead to irreversible sorption/swelling curves; indeed, if chemistry was involved in sorption-induced swelling, we would not recover the material properties when decreasing the moisture content down to $m = 0$ ($RH = 0$). In contrast, in agreement with our molecular simulation data, all experiments on sorption-induced swelling of wood/cellulose show that one recovers the dry state upon dehydrating samples that have been first hydrated. In order to exclude the sorption-related scenarios described above, we have conducted simulations of sorption in an undeformable host with or without relaxing the internal structure of cellulose. More precisely, in order to test the possibility that swelling hysteresis in cellulose pertains to adsorption alone, we have conducted the two following tests: we have simulated water adsorption/desorption in cellulose taken in its dry (unswollen) and fully hydrated (swollen) configurations. For these two states, we have performed two types of molecular simulations at constant volume: simulations in which the cellulose chains are allowed to relax and simulations in which the cellulose chains are kept frozen (i.e. no internal structure relaxation as the corresponding degrees of freedom are frozen). As shown in the figure below, the different datasets show that adsorption/desorption isotherms are hysteresis free, therefore supporting the picture that hysteresis in sorption-induced swelling of cellulose arises from the coupling between deformation and sorption (so that the free energy to be considered is a hybrid thermodynamic potential corresponding to the sum of the free energy of the cellulose host and the grand free energy of the confined water molecules).

Figure | Water adsorption and desorption in cellulose. Water sorption isotherms at $T = 300$ K in cellulose: (red circles) hybrid GCMC/MD molecular simulation, (black circles) sorption experiment from Ref.⁹. The experimental data were shifted up by $m = +0.05$ to account for the presence of residual, i.e. non-desorbable water, in the dried material (see text). Open and closed symbols are adsorption and desorption data, respectively. The adsorbed amount is expressed as a moisture content m defined as the mass of water per mass of dry material. The grey dashed lines correspond to the simulated, hysteresis-free sorption isotherms for water in the frozen unswollen and swollen cellulose material. The black dashed lines correspond to the same data but using simulations in which relaxation of the cellulose chains is allowed.

In order to address this important comment from Reviewer #2, we have added/replaced the figure reported above in the main manuscript. We have also included the following discussion in the revised version of the manuscript (Pages 7-8): “In order to probe the microscopic origin of hysteresis in sorption/swelling of cellulose, we performed additional simulations to test whether water adsorption/desorption cycles in undeformable cellulose lead to hysteresis. More precisely, we have simulated water adsorption/desorption in cellulose at taken in its dry (unswollen) and fully hydrated (swollen) configurations. For these two states, we have performed a set of molecular simulations at constant volume in which the cellulose chains are allowed to relax (flexible) and a set of molecular simulations at constant volume in which the cellulose chains are kept frozen (frozen). Figure 2 shows the sorption isotherms obtained by simulating water ad- and desorption in cellulose when the model is maintained in its unswollen dry

state and its swollen state at $RH = 1$. Adsorption/desorption cycles in the swollen and unswollen configurations are found to be hysteresis free (both in the flexible and frozen cellulose hosts). As expected, owing to the larger porous volume in swollen cellulose, the water adsorbed amount in the swollen state is larger than in the unswollen state for all RH . The different results above support the hypothesis that hysteretic behavior in sorption/swelling is governed by the coupling between sorption and swelling and not by a sorption or mechanical effect alone.”

We have also added the following discussion in the revised version of the manuscript (Page 7): “Both the simulation and experimental data exhibit significant hysteretic behavior upon increasing/decreasing RH . Before addressing the microscopic origin of sorption-induced swelling hysteresis, it should be noted that possible mechanisms relying on chemistry (i.e. involving bond formation/breaking) cannot be envisaged as they would lead to irreversible sorption/swelling curves; indeed, if chemistry was involved in sorption-induced swelling, one would not recover the material properties upon decreasing the moisture content down to $m = 0$ ($RH = 0$). In contrast, in agreement with our molecular simulation data, all experiments on sorption-induced swelling of wood/cellulose show that one recovers the dry state upon dehydrating samples that have been first hydrated.”

Comment 2. *I do not question the reliability of the simulation, but it is well-known that the accuracy of the classical force field is a big problem. For the hybrid strategy combining GCMC and MD, has it been verified? Is there any validation?*

Reply 2. We agree with the reviewer that our initial manuscript did not discuss carefully the validity of the forcefield used to describe cellulose (note that this point was already addressed in our reply to Comment #2 by Reviewer #1). In order to answer this important point, two modifications have been made:

- We have added in the revised version of our Supplementary Information a section dedicated to the discussion on the forcefield validity (SI file, Page 3):

“II. Discussion on the forcefield validity to describe cellulose

PCFF is a force field parameterized against a broad range of experimental observables for organic compounds. It has been applied to modelling cellulose-based materials by many researchers who have found that it captures quantitatively or semi-quantitatively most physical properties. More in details, for instance, Chen et al. [2] built an amorphous cellulose model and showed that the final density obtained is consistent with its experimental counterpart (1.39 g/cm^3 versus 1.48 g/cm^3). Similarly, in the present work, we found that the final density of our models (ranging from 1.39 to 1.41 g/cm^3) is consistent with typical experimental densities for cellulose (see Supplementary Table 1). As far as

mechanical properties are concerned, Tanaka and Iwata [3] used molecular simulation with the same forcefield to assess Young's Modulus of cellulose crystals; these authors found values in the range 124-155 GPa that are in good agreement with the experimental value (~138 GPa) [4] (no experimental mechanical data are available for amorphous cellulose). As for adsorption properties, in their molecular simulation of adsorption onto cellulose, Perez et al. [5] found that the heat of adsorption for a large variety of aromatic compounds is consistent with their experimental counterpart (84% of the adsorbate-cellulose couples displayed differences < 20% between the measured and predicted heats of adsorption). Xu and Chen [6] also found that PCFF predicts formaldehyde diffusion in cellulose with a temperature dependence of the self-diffusion coefficient in good agreement with the experimental data. Finally, in the context of the present work on water adsorption/desorption in cellulose, we emphasize that PCFF leads to cellulose/water hydrogen bonds with a typical energy (5.4 kcal/mol, see discussion in our manuscript) that is consistent with the conformational analysis made by Pizzi et al. [7,8]; these authors estimated theoretically that the sorption energy is around 5.5 kcal/mol for cellulose I crystals and 6.5 kcal/mol for paracrystalline (amorphous) cellulose. Overall, the discussion above shows that the forcefield used in the present work provides a reasonable, at least semi-quantitative, description of cellulose (including its density, mechanical, and adsorption properties)."

- For the sake of clarity, we have also added the following sentence in the revised version of the manuscript (Page 4):

"The parameters were taken from the PCFF forcefield²⁴ (see Section 2 in the Supplementary Information for a discussion on the validity of the forcefield to describe cellulose)."

Comment 3. *Concerning the results and their interpretation, some treatments are clearly lack of strong proof. For example, lines 138-139, "For the sake of comparison, the experimental data were shifted up by $m = +0.05$ to account for the presence of residual i.e. non-desorbable water in the 'dried' sample." It sounds reasonable but I rather believe that it should be much more complex. Why a shift of 0.05, not any other possible value? Is the presence of residual the only reason resulting the difference?*

Reply 3. We agree with the reviewer that this point should be discussed more carefully. From an experimental viewpoint, cellulose samples are difficult to totally dehydrate prior to water adsorption experiments so that they are thought to contain a residual number of undesorbed water molecules. Even when exposing samples to a relative humidity $RH = 0$, it would take an unreachable time to remove the last trapped water molecules. We mention that increasing the temperature to accelerate the removal of bonded water is not an option for temperature-sensitive materials such as cellulose. In addition to the uncertainty over this initial moisture content, such measurements (which include those by Mihranian et al that are used here) as a function of the relative humidity suffer from the following limitation; the sample is first placed in a desiccator that contains a recipient filled with salty water at a given salt

concentration to adjust the relative humidity (for $RH = 0$, a very good desiccant is used instead of a water-filled recipient). Even when sufficient time would allow the sample to reach thermodynamic equilibrium, measurement of the moisture content requires to take the sample out of the humidity chamber so that it can be weighed (the moisture content is simply obtained by subtracting the initial mass of the cellulose sample). While this operation is done as fast as possible, it cannot be ruled out that partial rehydration occurs since the air is not dry in lab conditions. Coming back to the comparison between the simulated and experimental data, while there is no formal justification about the exact shift to be used, we emphasize that a shift $m = +0.05$ allows getting good agreement between the two datasets. In particular, when corrected for such a shift, the experimental data not only show typical moisture contents - including the maximum moisture content at $RH = 1$ - but also slopes in the adsorption/desorption isotherm that are fully consistent with the simulated data. In order to address this comment, we have added the following discussion in our revised manuscript (Pages 6-7):

“Experimentally, cellulose is difficult to obtain totally dehydrated prior to such water adsorption experiments so that they contain a residual number of undesorbed water molecules. Moreover, such measurements as a function of RH suffer from the following limitation⁹. Even if sufficient time was allowed to reach thermodynamic equilibrium (including at $RH = 0$), measurement of the moisture content requires to take the sample out of the equilibration chamber so that it can be weighed. While this operation is done as fast as possible, some partial rehydration occurs. Consequently, for the sake of comparison, the experimental data were shifted up by $m = +0.05$ to account for the presence of residual i.e. non-desorbable water in the ‘dried’ sample and unavoidable partial rehydration upon weighing. While there is no formal justification for the exact shift used, the shift $m = +0.05$ allows getting good agreement between the two datasets. In particular, when corrected for such a shift, the experimental data show typical moisture contents – including the maximum moisture content at $RH = 1$ – and slopes in the adsorption/desorption isotherm that are consistent with the simulated data.”

Minor point 1. *Terminology of MD simulations is not consistent through the entire manuscript, such as isothermal-isobaric ensemble (MD-N σ T) in line 115 [then σ was used as the size of the confined molecule (line 161)], however in the Methods section, it becomes to “constant temperature/pressure with a Nose-Hoover thermostat/barostat (NPT)” in line 353.*

Reply 1. We apologize for the lack of consistent notations. We have checked carefully the revised manuscript and Supporting Information to check that all notations are now consistent. In the revised version, σ is only used to define the external stress applied to the sample while λ is used to refer to the size of the confined (water) molecule.

Minor point 2. *I am not sure whether all the simulation can be reproduced with the given poor description, especially for the GCMC part, regarding to change the RH.*

Reply 1. We agree with the reviewer that essential details were missing in our initial submission (this

point was also raised by Reviewer #1). In our reply to the same comment by Reviewer #1 (see our response to his/her comment #3), we have added a detailed paragraph in the Supplementary Information which we paste here for the sake of convenience (Page 2):

“I. Combining Grand Canonical Monte Carlo and Molecular Dynamics

As explained in our paper, state-of-the-art techniques such as Molecular Dynamics (MD) and Grand Canonical Monte Carlo (GCMC) do not allow describing coupled adsorption/swelling in porous materials. Indeed, while the former requires to work with a constant number of molecules, the latter only applies to systems having a constant volume. Following previous works, including the work by Ghoufi and Maurin for adsorption in Metal Organic Frameworks [1], we use a hybrid molecular simulation that consists of combining GCMC and MD to perform simulations in the so-called osmotic statistical ensemble $\mu\sigma T$ where μ is the chemical potential of water, σ is the stress applied to the cellulose material, and T the temperature. To do so, part of the Markov chain used in the Grand Canonical Monte Carlo simulations is replaced by an MD trajectory at constant number of water molecules N , constant stress σ and temperature T . More in details, this means that, in addition to conventional MC steps in the GCMC algorithm (insertions and deletions), MD timesteps *i.e.* at constant number of molecules (allowing for water molecule translations or rotations and cellulose local relaxation at constant external stress) are added as the MD technique is more efficient at converging towards local equilibrium. Moreover, by using MD simulations in the isostress and isothermal ensemble, the host (hydrated) cellulose material is allowed to swell or shrink. As shown in previous works, such a hybrid strategy succeeds in capturing coupled adsorption/swelling phenomena including adsorption-induced phase transitions in breathing materials such as Metal Organic Frameworks [1].

In practice, our hybrid molecular simulations were performed at $T = 300$ K using a Berendsen thermostat and an anisotropic external stress $\sigma = 0$ Pa (with the following relaxation times: $\tau_T = 0.1$ ps and $\tau_s = 1$ ps). Water molecules are described using the SPC/E water model with the SHAKE algorithm to maintain its internal structure rigid. The MD trajectory was integrated using the velocity Verlet integrator with a timestep equal to 1 fs. Hybrid MD/GCMC molecular simulations consisted of performing a large number of blocks where one block corresponds to 2000 GCMC insertion/deletion attempts followed by 200 MD timesteps. In total, 10^5 blocks were first performed to equilibrate the system followed by 2×10^4 additional blocks to accumulate statistics.”

As for the specific comment regarding the change in RH, we have rephrased the following sentences

(Page 5 of the revised manuscript): “Since T is well below the critical point for water, its vapor is assumed to behave as an ideal gas $\mu \sim k_B T \ln P$ where P is the vapor pressure, T is the temperature and k_B is Boltzmann’s constant. In practice, this means that, to simulate a given relative humidity $RH = P/P_0$ where $P_0 = 1017$ Pa is the bulk saturation vapor pressure for the SPC/E water model at $T = 300$ K, one imposes in the hybrid GCMC/MD simulations a chemical potential $\mu - \mu_0 \sim k_B T \ln RH$ (μ_0 is the chemical potential of the water model at the bulk saturating vapor pressure P_0).”

Reviewers' comments:

Reviewer #1 (Remarks to the Author):

The authors have made proper additions/corrections to the manuscripts fully satisfying recommendations by this reviewer.

Reviewer #2 (Remarks to the Author):

I appreciate the authors' efforts to address all my concerns in detail. I believe that the authors also share the same thought with me that the revised manuscript has been significantly improved. However, before I can recommend its publication on Nat Commun, I still have the following comments and suggestions.

1. We know that in experiments, we take the vapor pressure of water to be 2.3 kPa. In the present work, the bulk saturation vapor pressure for the SPC/E water model at $T = 300$ K, 1017 Pa was used in the simulations. For me, such difference between experiment and simulation is acceptable. However, the authors then compared their simulation results with experimental work in Figure 2. Due to the fact, the vapor pressure should be different between simulations and experiments, even for the same RH. Thus, how can they be compared and show such good agreement?
2. To maintain constant pressure in MD simulations should not be an easy task as the temperature control, since the fluctuation of pressure should be very remarkable. How would this factor affect the implement of combining GCMC and MD.
3. For the dense system shown in Fig. 1 and Fig. S1, is it easy to insert water molecules during GCMC simulations?

Dr Benoit Coasne
Tel: +33 6 70 80 12 34
benoit.coasne@univ-grenoble-alpes.fr

Laboratoire Interdisciplinaire de Physique
140 Av. de la physique
38000 Grenoble

Grenoble, June 5, 2018

* * * * *

Reviewer #2

I appreciate the authors' efforts to address all my concerns in detail. I believe that the authors also share the same thought with me that the revised manuscript has been significantly improved. However, before I can recommend its publication on Nat Commun, I still have the following comments and suggestions.

Comment 1. *We know that in experiments, we take the vapor pressure of water to be 2.3 kPa. In the present work, the bulk saturation vapor pressure for the SPC/E water model at $T = 300$ K, 1017 Pa was used in the simulations. For me, such difference between experiment and simulation is acceptable. However, the authors then compared their simulation results with experimental work in Figure 2. Due to the fact, the vapor pressure should be different between simulations and experiments, even for the same RH. Thus, how can they be compared and show such good agreement?*

Reply 1. This is an interesting point raised by the reviewer. There is indeed a non negligible mismatch between the experimental saturating vapor pressure for water and its numerical counterpart when using the SPC/E water model. This raises the question whether adsorption isotherms should be compared when plotted as a function of reduced pressure P/P_0 or absolute pressure P . Going back to the fundamental definition of adsorption, the thermodynamic parameter that governs the adsorbed amount is the chemical potential difference $\Delta\mu = \mu - \mu_0$ with respect to the chemical potential at saturation μ_0 . Indeed, at the saturation chemical potential i.e. at the bulk gas/liquid coexistence, adsorption from a relative humidity $RH = 1$ ($P = P_0$) is maximum (any adsorption above μ_0 corresponds to intrusion and is therefore only relevant to hydrophobic materials). As a result, when comparing adsorption isotherms in different fluids, the natural choice is to use the chemical potential difference $\Delta\mu = \mu - \mu_0$. This is relevant when comparing different fluids but also different models for the same fluid when using molecular simulation (or as in the present paper when comparing

experimental and simulation data). Considering that water vapor at room temperature behaves as an ideal gas, we used the following relationship $RH = P/P_0 = \exp[\Delta\mu/k_B T]$. Following the reviewer's comment, we feel that this point should be discussed in the revised version of our manuscript. We have therefore added the following discussion in the revised version of the manuscript (SI file, Page 5): "There is a non negligible mismatch between the experimental saturating vapor pressure for water and its numerical counterpart when using the SPC/E water model. This raises the question whether adsorption isotherms should be compared when plotted as a function of reduced pressure P/P_0 or absolute pressure P . Considering the fundamental definition of adsorption phenomena (in terms of Gibbs adsorption equation or Polanyi theory for instance), the thermodynamic parameter that governs the adsorbed amount is the chemical potential difference $\Delta\mu = \mu - \mu_0$ with respect to the chemical potential at saturation μ_0 . Indeed, at the saturation chemical potential i.e. at the bulk gas/liquid coexistence, adsorption from a relative humidity $RH = 1$ ($P = P_0$) is maximum (any adsorption above μ_0 corresponds to intrusion and is therefore only relevant to hydrophobic materials). As a result, when comparing adsorption isotherms for different fluids, the natural choice is to use the chemical potential difference $\Delta\mu = \mu - \mu_0$. This is relevant when comparing different fluids but also different models for the same fluid when using molecular simulation (or as in the present paper when comparing experimental and simulation data). Considering that water vapor at room temperature behaves as an ideal gas, we used the following relationship $RH = P/P_0 = \exp[\Delta\mu/k_B T]$."

Comment 2. *To maintain constant pressure in MD simulations should not be an easy task as the temperature control, since the fluctuation of pressure should be very remarkable. How would this factor affect the implement of combining GCMC and MD.*

Reply 2. We agree with the reviewer that this point should be addressed in details. We have added the following figure and discussion in the revised version of the manuscript (Pages 3-4): "In order to check the efficiency of the phase space sampling, we monitored the pressure and volume along the hybrid GCMC/N σ T simulations. Supplementary Fig. S1 shows the volume of the simulation box as a function of the number of time steps in the molecular dynamics simulation N_{MD} . We recall that, after every 200 MD timesteps, the simulation procedure also includes a GCMC segment that consists of 2000 MC steps. The insert in supplementary Fig. S1 shows for two different chemical potentials μ how the volume V

fluctuates around its equilibrium value in the course of a block consisting of a MD segment and a GCMC segment (of course, the volume does not change during the GCMC segment as only the energy and number of molecules are allowed to change). Overall, the data in Supplementary Fig. S1 show that volume sampling in such hybrid GCMC/N σ T simulations is efficient and that the volume reaches its final equilibrium value provided simulations are long enough (typically, at high loadings, at least 10^7 MD timesteps are required but shorter simulations are needed for low loadings). We also show in Supplementary Fig. S2 the stress monitored during a MD segment. As expected, owing to the system size, fluctuations over the stress are large but we checked that its average value is equal to the stress imposed in the simulations. We note that the stress variations seen in Supplementary Fig. S2 are small compared to the stiffness of the material, ~ 10 GPa (as also reflected by the small volume variations in Supplementary Fig. S1).”

Supplementary Figure S1. Volume in nm^3 as a function of MD steps as obtained from hybrid GCMC/MD simulations. The data obtained for a relative humidity $RH = 0.025$ (black data) and $RH = 1$ (red data) are shown. The insert, which shows a zoom corresponding to the end of the hybrid simulation, illustrates how the volume V fluctuates around its equilibrium value in the course of a block consisting of a MD segment and a GCMC segment (the volume does not change during the GCMC segment as only the energy and number of molecules are allowed to change).

Supplementary Figure S2. Stress fluctuations in the course of a MD block performed in a hybrid GCMC/MD simulation. The data obtained for a relative humidity $RH = 0.025$ (black data) and $RH = 1$ (red data) are shown.

Comment 3. For the dense system shown in Fig. 1 and Fig. S1, is it easy to insert water molecules during GCMC simulations?

Reply 3. Yes, we can easily insert water molecules as, in MD simulations, the system is allowed to swell and provide more room for additional molecules (although it is easier at low moisture content than at higher ones). We agree with the reviewer that this point should be addressed in details. We have added the following figure and discussion in the revised version of the manuscript (Page 4): “We also checked for two water chemical potentials that the number of water molecules converges to its final equilibrium value. Supplementary Fig. S3 shows the number of water molecules N_{water} as a function of the number of GCMC steps N_{GCMC} (as explained above, such hybrid simulations also include MD segments which are not shown here for the sake of clarity since the number of molecules is constant in MD).”

Supplementary Figure S3. Number of water molecules N_{water} as a function of GCMC steps as obtained from hybrid GCMC/MD simulations. The data obtained for a relative humidity $RH = 0.025$ (black data) and $RH = 1$ (red data) are shown (N_{water} in the course of the MD segments is not shown because it is not allowed to change in the $N\sigma T$ ensemble).

REVIEWERS' COMMENTS:

Reviewer #2 (Remarks to the Author):

The authors have addressed all my comments sufficiently and now I can recommend its publication.